# CsrA Positively and Directly Regulates the Expression of the *pdu*, *pocR*, and *eut* Genes Required for the Luminal Replication of *Salmonella* Typhimurium

Jessica Nava-Galeana,[a] Helen Yakhnin,[b] Paul Babitzke,[b] Víctor H. Bustamante[a]

ªDepartamento de Microbiología Molecular, Instituto de Biotecnología, Universidad Nacional Autónoma de México, Cuernavaca, Morelos, México
ᵇDepartment of Biochemistry and Molecular Biology, Center for RNA Molecular Biology, Pennsylvania State University, University Park, Pennsylvania, USA

**ABSTRACT** Enteric pathogens, such as *Salmonella*, have evolved to thrive in the inflamed gut. Genes located within the *Salmonella* pathogenicity island 1 (SPI-1) mediate the invasion of cells from the intestinal epithelium and the induction of an intestinal inflammatory response. Alternative electron acceptors become available in the inflamed gut and are utilized by *Salmonella* for luminal replication through the metabolism of propanediol and ethanolamine, using the enzymes encoded by the *pdu* and *eut* genes. The RNA-binding protein CsrA inhibits the expression of HilD, which is the central transcriptional regulator of the SPI-1 genes. Previous studies suggest that CsrA also regulates the expression of the *pdu* and *eut* genes, but the mechanism for this regulation is unknown. In this work, we show that CsrA positively regulates the *pdu* genes by binding to the *pocR* and *pduA* transcripts as well as the *eut* genes by binding to the *eutS* transcript. Furthermore, our results show that the SirA-CsrB/CsrC-CsrA regulatory cascade controls the expression of the *pdu* and *eut* genes mediated by PocR or EutR, which are the positive AraC-like transcriptional regulators for the *pdu* and *eut* genes, respectively. By oppositely regulating the expression of genes for invasion and for luminal replication, the SirA-CsrB/CsrC-CsrA regulatory cascade could be involved in the generation of two *Salmonella* populations that cooperate for intestinal colonization and transmission. Our study provides new insight into the regulatory mechanisms that govern *Salmonella* virulence.

**IMPORTANCE** The regulatory mechanisms that control the expression of virulence genes are essential for bacteria to infect hosts. *Salmonella* has developed diverse regulatory mechanisms to colonize the host gut. For instance, the SirA-CsrB/CsrC-CsrA regulatory cascade controls the expression of the SPI-1 genes, which are required for this bacterium to invade intestinal epithelium cells and for the induction of an intestinal inflammatory response. In this study, we determine the mechanisms by which the SirA-CsrB/CsrC-CsrA regulatory cascade controls the expression of the *pdu* and *eut* genes, which are necessary for the replication of *Salmonella* in the intestinal lumen. Thus, our data, together with the results of previous reports, indicate that the SirA-CsrB/CsrC-CsrA regulatory cascade has an important role in the intestinal colonization by *Salmonella*.

**KEYWORDS** Csr, *Salmonella*, regulation

Address correspondence to Víctor H. Bustamante, victor.bustamante@ibt.unam.mx.

The authors declare no conflict of interest.

*Salmonella enterica* serovar Typhimurium (*S.* Typhimurium) causes intestinal disease in humans and many animals (1). Two groups of genes are primarily responsible for mediating intestinal colonization by *S.* Typhimurium: the SPI-1 (*Salmonella* pathogenicity island 1) genes and the *pdu/eut* genes. The SPI-1 genes encode a type three secretion system (TTSS-1), several effector proteins, and transcriptional regulators, which mediate the invasion of intestinal epithelium cells as well as an intestinal inflammatory response (2, 3). The *pdu* and *eut* genes encode enzymes that provide *Salmonella* with the ability to grow

in the presence of propanediol and ethanolamine, respectively, which are nonfermentable carbon compounds that are metabolized in the lumen of the inflamed intestine via tetrathionate respiration (4–7). In laboratory conditions, the SPI-1 genes are expressed when *Salmonella* is grown in lysogeny broth (LB) (under SPI-1-inducing conditions) (8, 9). The *pdu* and *eut* genes are expressed in LB at low levels, and their expression is activated in the presence of propanediol or ethanolamine, respectively (10, 11).

A myriad of regulators control the expression of the SPI-1 genes, most of which act on HilD, which is an AraC-like transcriptional regulator that is encoded within SPI-1 that directly or indirectly activates the expression of the SPI-1 genes and other related genes that are located outside of SPI-1 (2, 3, 12). Among the regulators controlling HilD are the BarA/SirA two-component system (TCS) and the Csr system (12, 13). The BarA/SirA and Csr systems are present in numerous bacteria, in which they control a wide variety of cellular processes by acting in a regulatory cascade (14, 15). The BarA/SirA TCS activates the transcription of the *csrB* and *csrC* genes encoding the CsrB and CsrC (CsrB/C) small RNAs (sRNAs), which bind to the RNA-binding protein CsrA (16, 17). CsrA binds to sequences containing conserved GGA motifs, which are generally located within the loops of hairpin structures in RNAs (18, 19). CsrB and CsrC contain multiple CsrA-binding sites, and they therefore antagonize CsrA activity on target transcripts (20, 21). The most common CsrA-mediated regulatory mechanism involves CsrA binding to multiple sites in 5' RNA leader regions, one of which overlaps the Shine-Dalgarno (SD) sequence, thereby repressing translational initiation, which often leads to the degradation of mRNAs (15, 22–27). However, CsrA can also activate the expression of some genes either by binding to leader sequences in mRNAs and thereby preventing the formation of secondary structures that sequester the SD sequence or by protecting mRNAs from attack by RNases and thereby stabilizing transcripts (15, 28, 29). CsrA binds to the *hilD* mRNA and blocks its translation, while the BarA/SirA TCS activates the expression of the CsrB/C sRNAs that antagonize the effect of CsrA, thereby favoring the expression of *hilD* (13, 30).

The expression of the *pdu* and *eut* genes is positively controlled by the AraC-like transcriptional regulators PocR and EutR, respectively (10, 11, 31, 32). Additionally, global expression studies implicate the BarA/SirA and Csr systems in the regulation of the expression of the *pdu* and *eut* genes (33, 34). However, the mechanism for this regulation is unknown. In this study, we show that CsrA activates the expression of the *pdu* and *eut* genes by directly binding to the *pocR*, *pduA*, and *eutS* transcripts, while SirA-CsrB/C reduces the expression of these genes by counteracting the effect of CsrA. Our results indicate that the regulation of the *pdu* and *eut* genes by the SirA-CsrB/C-CsrA cascade requires the presence of the transcriptional regulators PocR and EutR, respectively.

## RESULTS

**SirA, CsrB/C, and CsrA regulate *pdu* and *eut* expression.** To examine the regulation of the *pdu* and *eut* genes by SirA, CsrB/C, and CsrA, we first constructed plasmids with *lacZ* translational fusions carrying the full-length intergenic regulatory region upstream and the first codons of the *pduA* or *eutS* genes (the *pduA* and *eutS* are the first genes of the *pdu* and *eut* operons, respectively) (Fig. 1). Then, the expression of the generated *pduA*'-'*lacZ* and *eutS*'-'*lacZ* translational fusions was quantified in the WT *S.* Typhimurium SL1344 strain and its isogenic Δ*sirA* and Δ*csrB* Δ*csrC* mutants, as well as in the WT strain overexpressing CsrA and in the Δ*sirA* mutant overexpressing SirA or CsrB. We did not analyze the effect of the Δ*csrA* mutant, as this strain presented a severe growth defect (13, 35). The bacterial strains were grown in LB at 37°C, which are conditions under which SirA, CsrB/C, and CsrA control the expression of the SPI-1 virulence genes (SPI-1-inducing conditions) (8, 9). The expression of both the *pduA*'-'*lacZ* and *eutS*'-'*lacZ* fusions was increased in the Δ*sirA* and Δ*csrB* Δ*csrC* mutants as well as in the WT strain overexpressing CsrA from the plasmid pK3-CsrA, compared with the WT strain either with or without an empty vector (Fig. 2). Conversely, the expression of

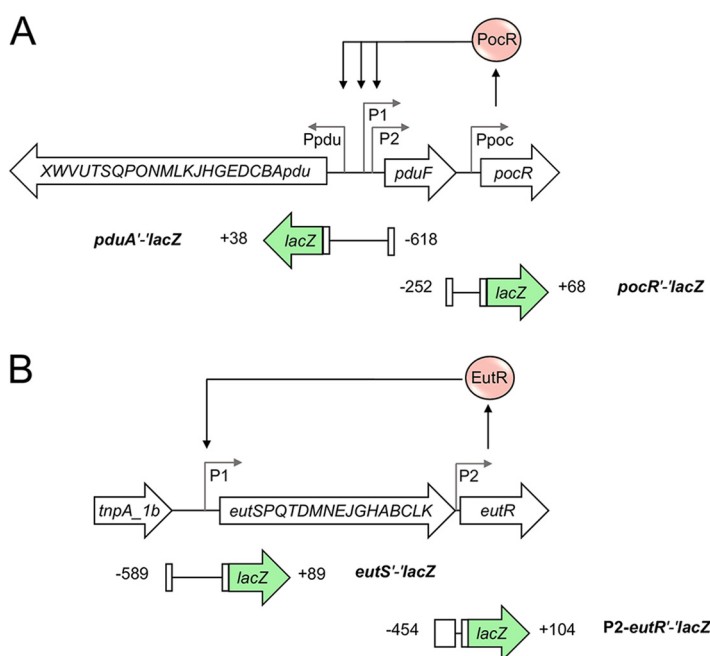

**FIG 1** Schematic representation of the *pdu* and *eut* operons. (A) The *pdu* and *pocR* genes. *pduA* is the first gene of the *pdu* operon. The PocR transcriptional regulator activates the expression of the *pdu* genes by acting on the promoters located upstream of *pduA* and *pduF* (adapted from [10]). (B) The *eut* genes. *eutS* is the first gene of the *eut* operon. The EutR transcriptional regulator activates the expression of the *eut* genes by acting on the promoter located upstream of *eutS* (adapted from [73]). The DNA fragments carried by the *pduA'*-'*lacZ* and *pocR'*-'*lacZ* fusions (A) as well as by the *eutS'*-'*lacZ* and *P2-eutR'*-'*lacZ* fusions (B) are shown; positions indicated are relative to the translational start codon.

both fusions was decreased in the Δ*sirA* mutant overexpressing SirA or CsrB from the plasmids pK3-SirA and pK3-CsrB, respectively, with respect to the Δ*sirA* mutant with or without an empty vector (Fig. 2). These results indicate that SirA and CsrB/C negatively control the expression of the *pdu* and *eut* genes, whereas CsrA positively controls their expression.

**The regulation of *pdu* and *eut* by SirA-CsrB/C-CsrA requires the presence of PocR or EutR.** The expression of the *pdu* and *eut* genes is positively controlled by the transcriptional regulators PocR and EutR, respectively (10, 11, 31, 32). Therefore, we asked whether the regulation of *pdu* and *eut* by the SirA-CsrB/C-CsrA cascade involves PocR or EutR. To test this possibility, we examined expression of the *pduA'*-'*lacZ* and *eutS'*-'*lacZ* fusions in the Δ*sirA* Δ*pocR*, Δ*csrB* Δ*csrC* Δ*pocR*, Δ*sirA* Δ*eutR*, and Δ*csrB* Δ*csrC* *eutR* mutant strains as well as in strains overexpressing CsrA (pK3-CsrA) in the presence or absence of PocR or EutR. The increased expression of *pduA'*-'*lacZ* and *eutS'*-'*lacZ* caused by the absence of SirA or CsrB/C or by the overexpression of CsrA (Fig. 2) was lost in the absence of the respective regulator PocR or EutR; the expression of the *pduA'*-'*lacZ* and *eutS'*-'*lacZ* fusions was barely detectable in the absence of PocR and EutR, respectively (Fig. 3). These results demonstrate that regulation of the *pdu* and *eut* genes by the SirA-CsrB/C-CsrA cascade requires the presence of PocR and EutR, respectively.

**SirA, CsrB/C, and CsrA regulate PocR and EutR expression.** Our results suggest that SirA-CsrB/C-CsrA regulates the *pdu* and *eut* genes through PocR and EutR. Therefore, we sought to determine whether SirA-CsrB/C-CsrA controls the expression of PocR and EutR.

The *pocR* gene is located near the *pdu* genes, and it is expressed as a single gene operon (Fig. 1A). We constructed a *pocR'*-'*lacZ* translational fusion carrying the full-length intergenic region upstream and the first codons of *pocR* (Fig. 1A), and we analyzed the expression of this translational fusion in the different *S.* Typhimurium strains that were assessed in this study. The expression of *pocR'*-'*lacZ* increased in the absence

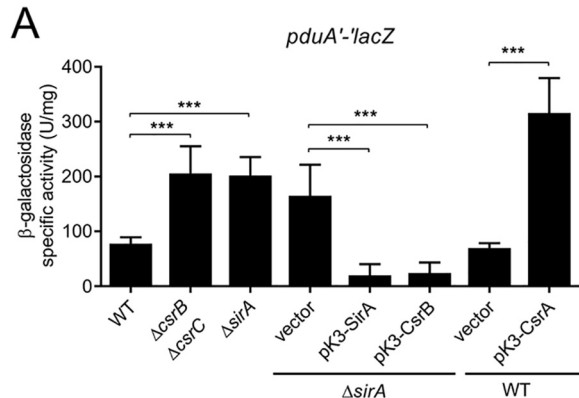

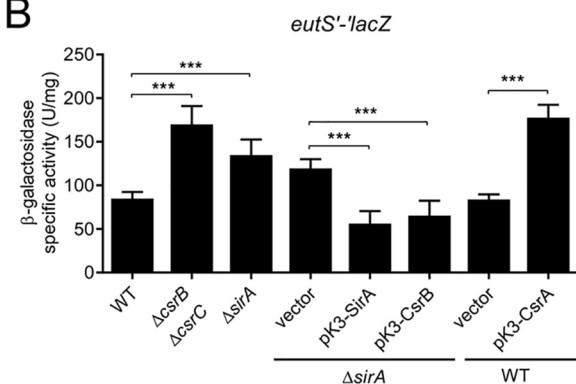

**FIG 2** The SirA-CsrB/C-CsrA cascade regulates the expression of the *pdu* and *eut* genes. The $\beta$-galactosidase activity of the *pduA'-'lacZ* (A) and *eutS'-'lacZ* (B) translational fusions was quantified in the indicated strains. $\beta$-galactosidase assays were performed with samples taken from bacterial cultures that were grown overnight in LB at 37℃. The data represent the average and the standard deviation of three independent experiments done in duplicate. The $P$ values were calculated using one-way ANOVAs with Tukey's *post hoc* tests (***, $P < 0.001$).

of SirA or CsrB/C and by the overexpression of CsrA from pK3-CsrA (Fig. 4A). Conversely, the expression of this fusion decreased by the expression of SirA or CsrB from pK3-SirA and pK3-CsrB, respectively (Fig. 4A). Consistent with these results, the production of CsrA from pK3-CsrA increased the chromosomal expression of *3xFLAG*-tagged PocR (PocR-FLAG) by 3.4-fold in the WT strain (Fig. 4B). These results show that the SirA-CsrB/C-CsrA cascade regulates the expression of *pocR*. To determine whether the presence of PocR is required for this regulation, we next examined the expression of the *pocR'-'lacZ* fusion in the Δ*sirA* Δ*pocR* and Δ*csrB* Δ*csrC* Δ*pocR* mutant strains as well as in strains overexpressing CsrA from pK3-CsrA in the WT and Δ*pocR* genetic backgrounds. The absence of SirA or CsrB/C as well as the overexpression of CsrA increased the expression of *pocR'-'lacZ* in the Δ*pocR* mutants (Fig. 5A), indicating that PocR is not required for the regulation of the *pocR* gene by the SirA-CsrB/C-CsrA cascade.

The *pocR'-'lacZ* fusion was expressed at similar levels in the WT strain and its iso-genic Δ*pocR* mutant when both were carrying an empty vector (Fig. 5A), indicating that PocR does not autoregulate its own expression. Consistent with this finding, elec-trophoretic mobility shift assays (EMSAs) showed MBP-PocR binding to the *pduA* promoter region but not to the *pocR* or *eutR* promoters (Fig. 5B and C). In agreement with these findings, previous studies indicated that PocR activates the expression of the *pdu* genes but does not regulate itself (10, 36). It should be noted that PocR binding to the *pduA* promoter had not been determined previously.

*eutR* is the last gene of the *eut* operon, and it is transcribed primarily from the pro-moter located upstream of *eutS*; however, an additional promoter upstream of *eutR*

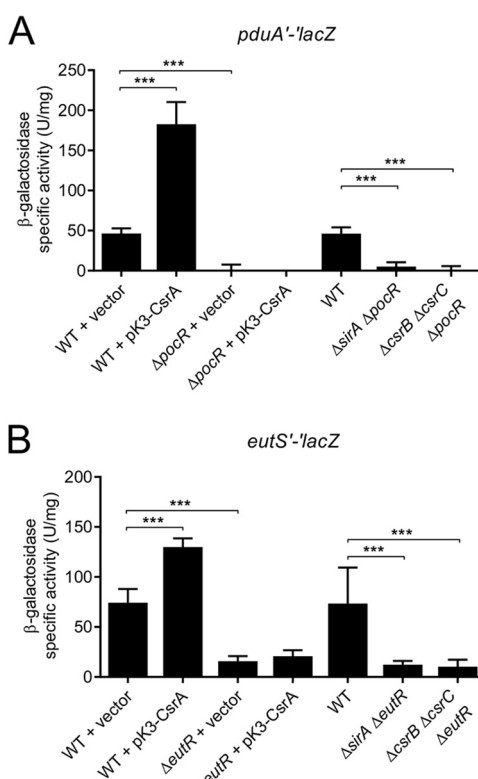

**FIG 3** The SirA-CsrB/C-CsrA cascade requires PocR and EutR to regulate the expression of the *pdu* and *eut* genes. The β-galactosidase activity of the *pduA'-'lacZ* (A) and *eutS'-'lacZ* (B) translational fusions was quantified in the indicated strains. β-galactosidase assays were performed with samples taken from bacterial cultures that were grown overnight in LB at 37°C. The data represent the average and the standard deviation of three independent experiments done in duplicate. The *P* values were calculated using one-way ANOVAs with Tukey's *post hoc* tests (***, $P < 0.001$).

has been reported (Fig. 1B) (11). Our results show that the SirA-CsrB/C-CsrA cascade controls the expression of *eutS* (Fig. 2B), implying that SirA-CsrB/C-CsrA would also regulate the expression of *eutR*. Consistent with this prediction, the production of CsrA from pK3-CsrA increased the chromosomal expression of *3xFLAG*-tagged EutR (EutR-FLAG) by 2.1-fold in the WT strain (Fig. 4D). To determine whether the SirA-CsrB/C-CsrA cascade also regulates the expression of a translational fusion driven by promoter P2 immediately upstream of *eutR* (Fig. 1B), we constructed and analyzed a P2-*eutR'-'lacZ* translational fusion. The absence of SirA or CsrB/C did not affect the expression of the P2-*eutR'-'lacZ* fusion (Fig. 4C). Intriguingly, the production of CsrA from pK3-CsrA significantly reduced the expression of the P2-*eutR'-'lacZ* fusion in the WT strain (Fig. 4C), which was an opposite effect to that observed for CsrA on *eutS* and EutR-FLAG (Fig. 2B, 3B, and 4D) as well as on *pduA*, *pocR*, and PocR-FLAG (Fig. 2A, 3A, 4A, 4B, and 5A).

Together, these results indicate that the SirA-CsrB/C-CsrA cascade regulates the expression of *pocR* by acting on the transcript generated by the promoter upstream of this gene and that it regulates the expression of *eutR* by acting on the transcript generated by the promoter upstream of *eutS*.

**SirA and CsrB/C regulate *pdu* and *eut* expression in the presence of inducer molecules.** The expression of the *pdu* and *eut* genes is activated to high levels in the presence of propanediol and ethanolamine, respectively; vitamin B$_{12}$ has also been shown to act as an inducer molecule for the expression of the *eut* genes (10, 11, 37). We found that SirA and CsrB/C also control the expression of the *pdu* and *eut* genes in the presence of inducer molecules (propanediol + vitamin B$_{12}$ or ethanolamine + vitamin B$_{12}$), but this is to a lower extent than that observed under SPI-1-inducing conditions (lacking inducers) (Fig. 6).

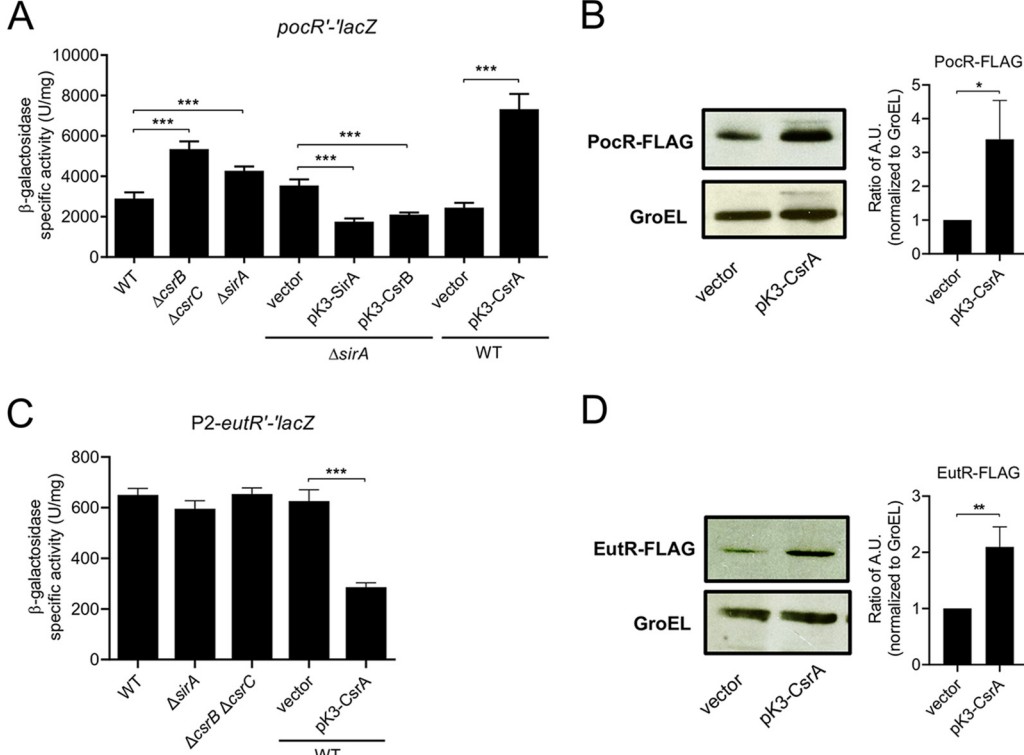

**FIG 4** The SirA-CsrB/C-CsrA cascade regulates the expression of PocR and EutR. The $\beta$-galactosidase activity of the *pocR'-'lacZ* (A) and P2-*eutR'-'lacZ* (C) translational fusions was quantified in the indicated strains. The data represent the average and the standard deviation of three independent experiments done in duplicate. The $P$ values were calculated using one-way ANOVAs with Tukey's *post hoc* tests (***, $P < 0.001$). Western blot analysis of PocR-FLAG (B) and EutR-FLAG (D) expression in the WT strain carrying a chromosomal *3xFLAG*-tagged *pocR* or *eutR* gene, respectively, with the indicated plasmids. Monoclonal anti-FLAG antibodies were used to detect the FLAG-tagged proteins. As a loading control, the expression of GroEL was also detected using polyclonal anti-GroEL antibodies. The blots were performed three times in independent experiments. Representative images of the blots are shown. The fold change for the expression of FLAG-tagged proteins were calculated as the ratio of AU (arbitrary units) normalized with GroEL, using the ImageJ software package. The $P$ values were obtained by using unpaired Student's *t* tests (*, $P < 0.05$; **, $P < 0.01$). Western blots and $\beta$-galactosidase assays were performed with samples taken from bacterial cultures that were grown overnight in LB at 37°C.

**CsrA binds to the *pocR*, *pduA*, and *eutS* transcripts.** To determine the direct targets of CsrA for the regulation analyzed in this study, we performed EMSAs using purified CsrA-H6 and the 5'-end-labeled RNA of *pocR*, *pduA*, and *eutS*. A band with lower mobility was detected with a CsrA-H6 concentration between 1 and 8 nM for *pocR* RNA, between 8 and 63 nM for *pduA* RNA, and between 2 and 16 nM for *eutS* RNA (Fig. 7), indicating that CsrA binds to these three transcripts. A nonlinear least-squares analysis of these data yielded apparent $K_d$ values of 7.4, 31, and 5.7 nM for *pocR*, *pduA*, and *eutS* RNAs, respectively (Fig. 7), indicating that CsrA has a higher affinity for *pocR* and *eutS* than for *pduA*. Additionally, the specificity of the CsrA interaction with *pocR*, *pduA*, and *eutS* RNAs was evaluated by competition experiments with specific (*pocR*, *pduA*, and *eutS*) and nonspecific (*phoB*) unlabeled RNA competitors. Unlabeled *pocR*, *pduA*, or *eutS* RNAs were effective competitors, whereas *phoB* RNA was not (Fig. 7), indicating that these interactions are specific. Collectively, our results show that CsrA positively regulates the expression of the *pdu* genes by binding to the *pocR* and *pduA* transcripts, and it also positively regulates the expression of the *eut* genes by binding to the *eutS* transcript.

## DISCUSSION

In the mouse intestine as well as under SPI-1-inducing *in vitro* conditions, *Salmonella* differentiates into two subpopulations that are genetically identical but phenotypically distinct: one subpopulation expresses SPI-1 genes (SPI-1[ON]), and the other does not (SPI-1[OFF])

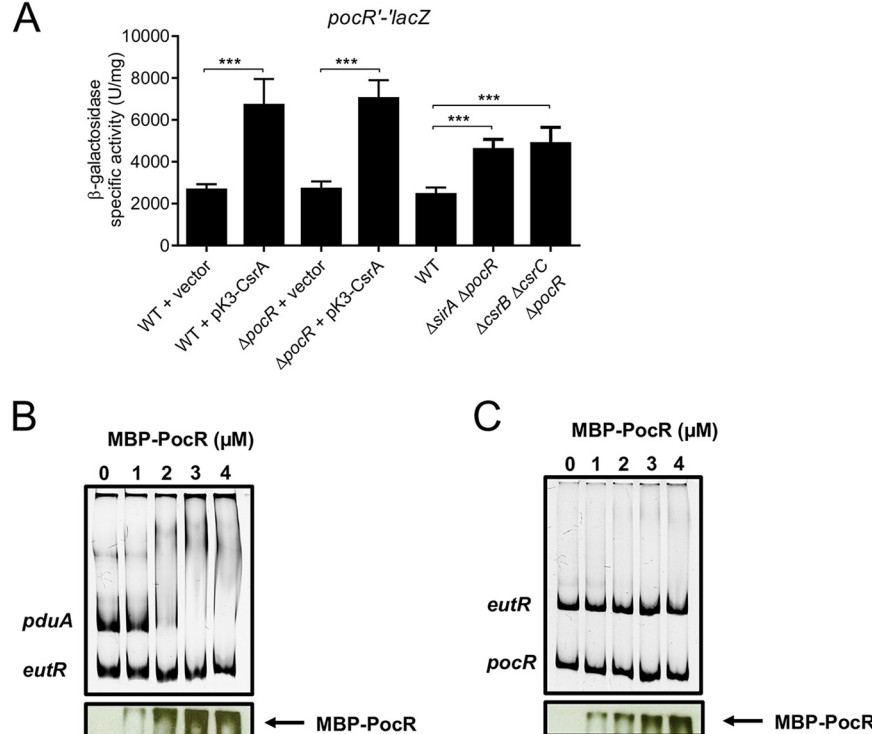

**FIG 5** The SirA-CsrB/C-CsrA cascade regulates the expression of *pocR* in the absence of PocR. (A) The β-galactosidase activity of the *pocR'-'lacZ* translational fusion was quantified in the indicated strains. β-galactosidase assays were performed with samples taken from bacterial cultures that were grown overnight in LB at 37℃. The data represent the average and the standard deviation of three independent experiments done in duplicate. The *P* values were calculated using one-way ANOVAs with Tukey's *post hoc* tests (***, $P < 0.001$). Nonradioactive EMSAs using purified MBP-PocR and the DNA fragments contained in the *pduA'-'lacZ* (B) and *pocR'-'lacZ* (C) fusions. As a negative control, the DNA fragment contained in the *eutR'-'lacZ* fusion was included in each DNA-binding reaction. The immunodetection assays using anti-MBP monoclonal antibodies (the images below the EMSAs) show that the MBP-PocR protein forms overly large complexes with (*pduA*) or without (*pocR* and *eutR*) bound DNA, and these remained near the wells of the gels.

(38–42). Different studies indicate that *Salmonella* colonizes the gut via a division of labor between these two subpopulations: the SPI-1$^{ON}$ is able to invade cells from the intestinal epithelium and trigger an inflammatory response that provides a particular niche where the SPI-1$^{OFF}$ replicates in the intestinal lumen, thereby displacing the microbiota (39–41, 43–45).

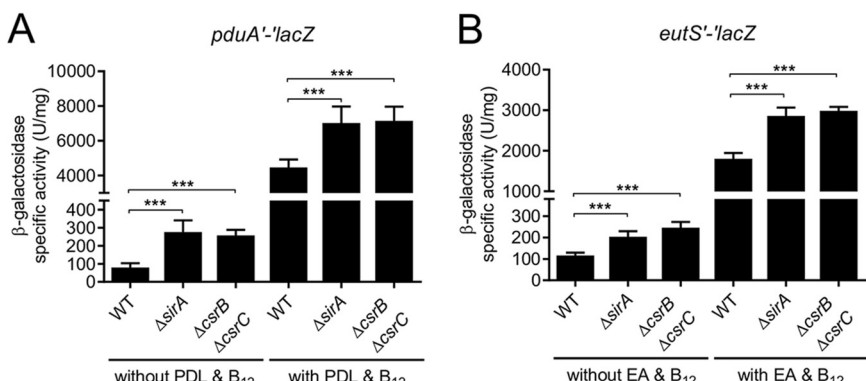

**FIG 6** SirA and CsrB/C regulate the expression of the *pdu* and *eut* genes in the absence or presence of inducer molecules. The β-galactosidase activity of the *pduA'-'lacZ* (A) and *eutS'-'lacZ* (B) translational fusions was determined in the indicated strains that were grown in the absence or presence of either 12.5 mM 1,2-propanediol and 150 nM vitamin B$_{12}$ (A) or 10 mM ethanolamine and vitamin B$_{12}$ (B). β-galactosidase assays were performed with samples taken from bacterial cultures that were grown overnight in LB at 37℃. The data represent the average and the standard deviation of three independent experiments done in duplicate. The *P* values were calculated using one-way ANOVAs with Tukey's *post hoc* tests (***, $P < 0.001$).

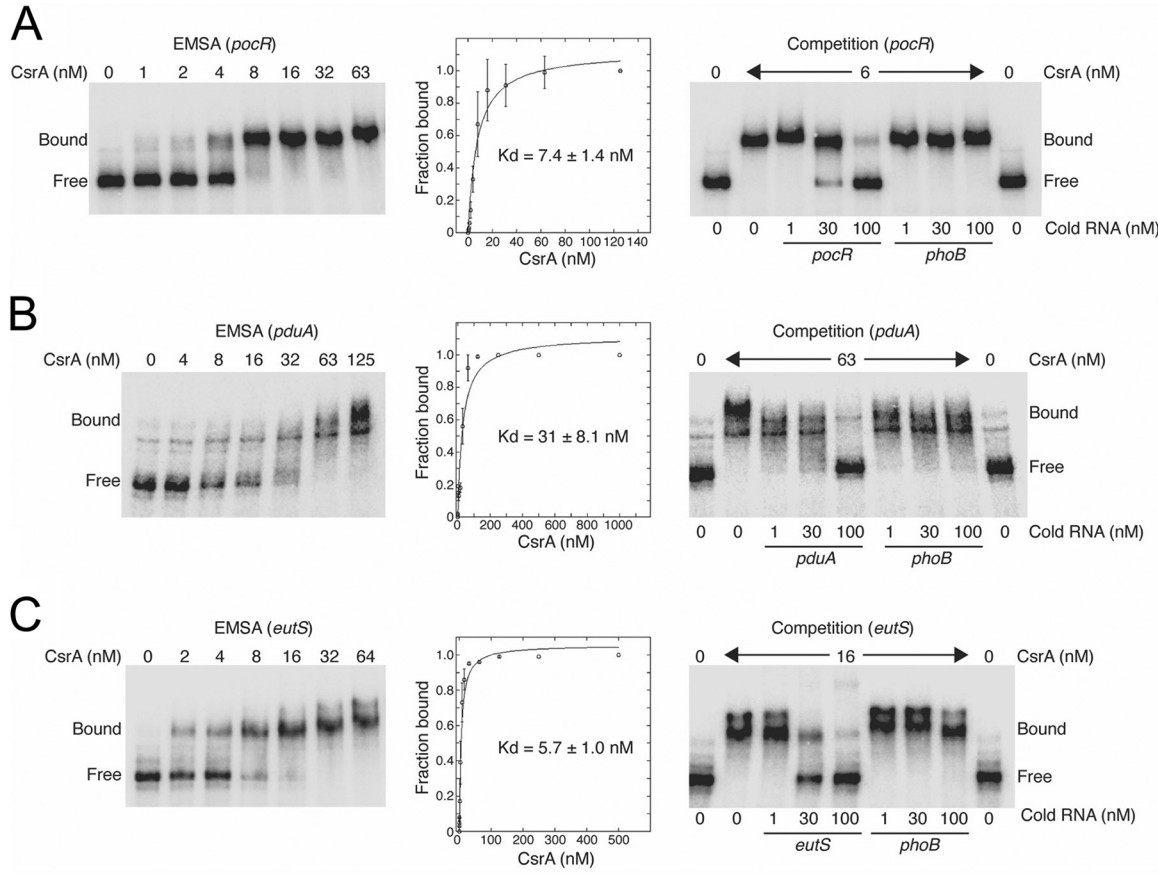

**FIG 7** CsrA binds specifically to the *pocR*, *pduA*, and *eutS* transcripts. CsrA binding to the *pocR* (A), *pduA* (B), and *eutS* (C) RNAs was analyzed using EMSAs by incubating $^{32}$P-labeled RNA fragments with increasing concentrations of purified CsrA-H6. The positions of bound and free RNA are marked. The simple binding curves for these data are shown. The $K_d$ values of the CsrA interaction with the *pocR*, *pduA*, and *eutS* transcripts are shown. For the RNA competition experiments, labeled *pocR* (0.1 nM), *pduA* (0.2 nM), or *eutS* (0.2 nM) RNA was combined with 1, 30, or 100 nM unlabeled specific (*pocR*, *pduA*, and *eutS*) or nonspecific (*phoB*) competitor RNA and was incubated with purified CsrA.

Alternative electron acceptors, such as tetrathionate (5), which is utilized by *Salmonella* for propanediol and ethanolamine metabolism (4–7, 46), are generated in the inflamed gut. The *pdu* and *eut* genes encode the enzymes necessary for the use of propanediol and ethanolamine as carbon sources (32, 47). The BarA/SirA-CsrB/C-CsrA regulatory cascade controls the expression of the SPI-1 genes under SPI-1-inducing conditions (13). BarA/SirA-CsrB/C induces the expression of these genes by antagonizing the translational repression exerted by CsrA on the *hilD* transcript (13, 30), which codes for the central positive regulator of SPI-1 (48–50). Recently, we reported that SirA is required for the generation of the SPI-1$^{ON}$ subpopulation (51). Results from the present study, together with the results of previous reports (33, 34), show that the BarA/SirA-CsrB/C-CsrA cascade also controls the expression of the *pdu* and *eut* genes under SPI-1-inducing conditions, but, interestingly, this occurs in a manner that is opposite to that of the control exerted by this regulatory cascade on the SPI-1 genes (Fig. 8). Our data indicate that SirA-CsrB/C negatively controls the *pdu* and *eut* genes by counteracting the direct positive regulation of CsrA on the *pocR-pduA* and *eutS* transcripts, respectively. We found that CsrA binds to the *pocR* and *eutS* RNAs with a similar affinity to that reported for its interaction with *hilD* RNA (13) and that the affinity of CsrA for *pduA* RNA was approximately 5-fold lower. CsrA primarily represses the translation of target mRNAs (13, 15, 22–27). However, positive regulation by CsrA on some transcripts has been described. For instance, CsrA positively controls the expression of the master regulator of flagellar genes, namely, FlhD$_4$C$_2$, by binding to the *flhDC* mRNA and thereby protecting it from degradation by RNase E (28). Furthermore, CsrA activates the expression of YmdA, which is a protein of unknown function, by binding to the *ymdA* mRNA and thereby exposing the SD sequence

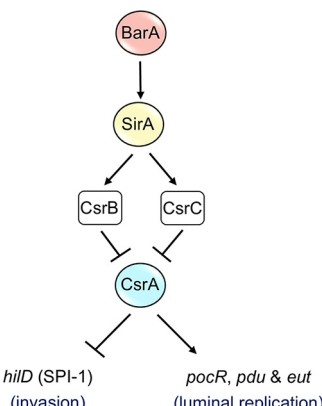

**FIG 8** The model for the regulation of the *pdu/pocR/eut* and *hilD* (SPI-1) genes by BarA/SirA-CsrB/C-CsrA. The expression of the *pdu/pocR/eut* and *hilD* (SPI-1) genes that are required for luminal replication and for the invasion of host cells, respectively, is oppositely controlled by the BarA/SirA-CsrB/C-CsrA regulatory cascade. See the text for details.

for translation initiation (29). The precise mechanism(s) by which CsrA activates the expression of *pocR*, *pduA*, and *eutS* remains to be determined.

Vitamin B$_{12}$ is necessary for the metabolism of propanediol and ethanolamine because it acts as a cofactor for the propanediol dehydratase enzyme that is encoded by the *pdu* genes as well as for the ethanolamine ammonia-lyase enzyme that is encoded by the *eut* genes (31, 32, 47, 52, 53). *Salmonella* anaerobically synthesizes vitamin B$_{12}$ with the enzymes encoded in the *cbi-cob* operon (36, 54, 55). In addition, the reduction of tetrathionate during the metabolism of propanediol and ethanolamine requires tetrathionate reductase, thiosulphate reductase, and sulfite reductase enzymes, which are encoded by the *ttr*, *phs*, and *asr* operons, respectively (4, 56). Interestingly, our unpublished results and previous reports support that the BarA/SirA-CsrB/C-CsrA regulatory cascade controls the expression of the *cbi-cob*, *phs*, and *asr* genes in a similar way to that observed for the *pdu* and *eut* genes (33, 34).

Based on the results described above, it is tempting to speculate that the BarA/SirA-CsrB/C-CsrA regulatory cascade could be involved in the generation of the SPI-1$^{ON}$ and SPI-1$^{OFF}$ subpopulations, which is a matter of our current investigation. Molecules or cues that are present in the intestine, acting through the BarA/SirA TCS and/or other regulatory pathways, could mediate whether the expression of SPI-1 or that of the *pdu/eut/cbi-cob* genes is induced. For instance, the short-chain fatty acids (SCFAs) acetate and formate act through the BarA/SirA TCS to activate SPI-1 gene expression (57–59). Other SCFAs or long-chain fatty acids (LCFAs), such as propionate, butyrate, oleate, myristate, and palmitate, repress the expression of the SPI-1 genes (57, 60–63). It is necessary to know how all of these molecules or signals acting on SPI-1 affect the expression of the *pdu/eut/cbi-cob* genes. This will help to integrate the possible pathways that oppositely control the expression of the SPI-1 and the *pdu/eut/cbi-cob* genes, which could be involved in the generation of the SPI-1$^{ON}$ and SPI-1$^{OFF}$ subpopulations. It should be noted that propanediol, which is the inducer molecule for *pdu* expression, indirectly represses SPI-1 via propionate production from propanediol metabolism (64). Our study adds an additional layer to the complex regulatory network that controls *Salmonella* virulence.

## MATERIALS AND METHODS

**Bacterial strains and growth conditions.** The bacterial strains used in this study are listed in Table 1. The cultures for the $\beta$-galactosidase and Western blot assays were grown in test tubes containing 5 mL of lysogeny broth (LB)-Miller (1% tryptone, 0.5% yeast agar, and 1% NaCl [pH 7.5]), that were incubated for 16 h at 37°C with shaking at 200 rpm. When necessary, the culture medium was supplemented with streptomycin (100 $\mu$g/mL), ampicillin (200 $\mu$g/mL), or kanamycin (30 $\mu$g/mL).

**Construction of mutants and strains expressing *3xFLAG*-tagged proteins.** Nonpolar gene deletion mutant strains were generated by the λRed recombinase system, as reported previously (65)

**TABLE 1** Bacterial strains and plasmids used in this study[a]

| Strain or plasmid | Genotype or description | Reference or source |
|---|---|---|
| Strain | | |
| *S.* Typhimurium | | |
| SL1344 | *xyl*, *hisG*, *rpsL*, Sm^R | 74 |
| JPTM27 | Δ*sirA* | 13 |
| DTM134 | Δ*csrB* Δ*csrC* | 51 |
| DTM240 | Δ*pocR::kan* | This study |
| DTM241 | Δ*pocR* | This study |
| DTM242 | Δ*eutR::kan* | This study |
| DTM243 | Δ*eutR* | This study |
| DTM244 | Δ*sirA* Δ*pocR::kan* | This study |
| DTM245 | Δ*csrB* Δ*csrC* Δ*pocR::kan* | This study |
| DTM246 | Δ*sirA* Δ*eutR::kan* | This study |
| DTM247 | Δ*csrB* Δ*csrC* Δ*eutR::kan* | This study |
| DTM248 | *pocR::3XFLAG-kan* | This study |
| DTM249 | *pocR::3XFLAG* | This study |
| DTM250 | *eutR::3XFLAG-kan* | This study |
| DTM251 | *eutR::3XFLAG* | This study |
| *E. coli* DH10β | Laboratory strain | Invitrogen |
| BL21/DE3 | Strain for the expression of recombinant proteins | Invitrogen |
| | | |
| Plasmids | | |
| pSUB11 | pG704 derivative template plasmid for FLAG epitope tagging | 66 |
| pKD46 | pINT-ts derivative containing a red recombinase system under an arabinose-inducible promoter | 65 |
| pCP20 | Plasmid expressing FLP recombinase from a temp-inducible promoter, Ap^R | 65 |
| pRS414 | pRS415 and pMC1403 derivative plasmid for *lacZ* translational fusions, Ap^R | 67 |
| ppduA-lacZ | pRS414 derivative containing *pduA'-'lacZ* translational fusion from nucleotides −618 to +38 | Nava-Galeana et al., submitted |
| peutS-lacZ | pRS414 derivative containing *eutS'-'lacZ* translational fusion from nucleotides −589 to +89 | This study |
| ppocR-lacZ | pRS414 derivative containing *pocR'-'lacZ* translational fusion from nucleotides −252 to +68 | This study |
| peutR-lacZ | pRS414 derivative containing P2-*eutR'-'lacZ* translational fusion from nucleotides −454 to +104 | This study |
| pMPM-K3 | p15A derivative low copy number cloning vector, *lac* promoter, Kan^R | 75 |
| pK3-CsrA | pMPM-K3 derivative expressing CsrA from the *lac* promoter | 13 |
| pK3-SirA | pMPM-K3 derivative expressing SirA from the *lac* promoter | 13 |
| pK3-CsrB | pMPM-K3 derivative expressing CsrB from the *lac* promoter | 13 |
| pMAL-c2x | Vector for constructing maltose binding protein (MBP) fusions, *lac* promoter, Ap^R | New England Biolabs |
| pMAL-PocR | pMAL-c2x derivative expressing MBP-PocR fusion protein, Ap^R | This study |

[a]The coordinates for the *lacZ* translational fusions are relative to the first base of the start codon for each gene. Sm^R, streptomycin resistance; Ap^R, ampicillin resistance; Kan^R, kanamycin resistance.

(Table 1). The *pocR* and *eutR* genes were replaced with a selectable kanamycin resistance cassette in the *S.* Typhimurium SL1344 strain, thereby generating the Δ*pocR::kan* (DTM240) and Δ*eutR::kan* (DTM242) mutants, respectively. The Δ*sirA* Δ*pocR::kan* (DTM244), Δ*csrB* Δ*csrC* Δ*pocR::kan* (DTM245), Δ*sirA* Δ*eutR::kan* (DTM246), and Δ*csrB* Δ*csrC* Δ*eutR::kan* (DTM247) double and triple mutants were generated by introducing the Δ*pocR::kan* (DTM240) or the Δ*eutR::kan* (DTM242) allele into the Δ*sirA* (JPTM27) or Δ*csrB* Δ*csrC* (DTM134) mutants via P22 transduction. The chromosomal *pocR* and *eutR* genes were *3xFLAG*-tagged in the *S.* Typhimurium SL1344 strain by using the previously reported λRed recombinase system (66), thereby generating the *pocR::3xFLAG-kan* (DTM248) and *eutR::3xFLAG-kan* (DTM250) strains, respectively. When required, the kanamycin resistance cassette was excised from the respective mutant strain by using the helper plasmid pCP20 expressing the FLP recombinase, as described previously (65), thereby generating the *S.* Typhimurium Δ*pocR* (DTM241), Δ*eutR* (DTM243), *pocR::3xFLAG* (DTM249), and *eutR::3xFLAG* (DTM251) strains. All of the mutant strains were verified via PCR amplification and sequencing.

**Construction of plasmids.** The plasmids and primers used in this study are listed in Tables 1 and 2, respectively. To construct the peutS-*lacZ*, ppocR-*lacZ*, and peutR-*lacZ* plasmids, the regulatory regions of *eutS*, *pocR*, and *eutR* were amplified by PCR using the primer pairs eutS-FwEcoRI/R2eutS-BamHI, pocR-FwEcoRI/pocR-RvBamHI and F2eutR-EcoRI/R2eutR-BamHI, respectively. The PCR products were digested with EcoRI and BamHI and were then cloned into the pRS414 vector (67) digested with the same enzymes. To construct the pMAL-PocR plasmid expressing the MBP-PocR fusion protein, the *pocR* structural gene was amplified by PCR using the primers F2PocR-MBP and R2PocR-MBP. The PCR product was digested with BamHI and PstI and was cloned into vector pMAL-c2x digested with the same enzymes. Chromosomal DNA from the WT *S.* Typhimurium SL1344 strain was used as the DNA template in all of the PCRs. All of the plasmids were sequenced and then transformed via electroporation into *S.* Typhimurium genetic backgrounds, as specified.

**β-galactosidase assay.** The protein quantification and β-galactosidase activity measurements were performed as previously described (68, 69), with the following modifications. Samples of cells were

**TABLE 2** Primers used in this work

| Primer | Sequence (5′–3′)$^{a,b}$ | Target gene | RE$^c$ |
|---|---|---|---|
| For *lacZ*-translational fusions | | | |
| eutS-FwEcoRI | CTT*GAATTC*GAACACGGCGAAGATACAGG | *eutS* | EcoRI |
| R2eutS-BamHI | CTT*GGATCC*GCCAGTTCTTCACCAGGGTG | *eutS* | BamHI |
| pocR-FwEcoRI | CTT*GAATTC*GGTAAGAATTTACCTTGTAACC | *pocR* | EcoRI |
| pocR-RvBamHI | CTT*GGATCC*GTGGCTTGTGCAAAATCCTG | *pocR* | BamHI |
| F2eutR-EcoRI | CTT*GAATTC*GTTTGCTCAGTCATCAAGTGC | *eutR* | EcoRI |
| R2eutR-BamHI | CTT*GGATCC*CGCTGATGAACATTGTCCACC | *eutR* | BamHI |
| | | | |
| For gene deletion | | | |
| pocR-H1P1 | TAAATTAACTGAGGGGTTTTATCATGATTTCTGCGAGCGCTC<u>TGTAGGCTGGAGCTGCTTCG</u> | *pocR* | No RE |
| eutR-H1P1 | GCATAGAAGATCATGAAAAAGACCCGTACAGCGAATTTGCAC<u>TGTAGGCTGGAGCTGCTTCG</u> | *eutR* | No RE |
| | | | |
| For gene FLAG-tagging | | | |
| pocR-FLAGFw | TATCGCCAGCAGATAAATGAGAATTCTCATCCTCCATCGTTA<u>GACTACAAAGACCATGACGG</u> | *pocR* | No RE |
| pocR-FLAGRv | ACAAAAGACTATCAAAAATCGGCAATAGCAAAATATTGCTAT<u>CATATGAATATCCTCCTTAG</u> | *pocR* | No RE |
| eutR-FLAGFw | AAACCGTCGTTGACGCTGCATCAACGGATGCGGCAATGGGCT<u>GACTACAAAGACCATGACGG</u> | *eutR* | No RE |
| eutR-FLAGRv | CACGCGCACGTTATCAGCAACCGGAGAGCCTCCCCATCAATA<u>CATATGAATATCCTCCTTAG</u> | *eutR* | No RE |
| | | | |
| For gene cloning | | | |
| F2PocR-MBP | CTT *GGA TCC* ATT TCT GCG AGCGCT CTG AAC TC | *pocR* | BamHI |
| R2PocR-MBP | CTT *CTG CAG* GAC TAT CAA AAATCG GCA ATA GC | *pocR* | PstI |
| | | | |
| For RNA EMSAs | | | |
| T7P *pduA* For | CTAATACGACTCACTATAGGGGTTCTTATAGTCCCAACTATCGGAACACTCC | *pduA* | No RE |
| *pduA* Rev | CCTTTGGTTTCTACCATTCCTAGTGCTTC | *pduA* | No RE |
| T7P *pocR* For | CTAATACGACTCACTATAGGGACTTTTTATCAGGGCCAGGATAATGG | *pocR* | No RE |
| *pocR* Rev | CATGATAAAACCCCTCAGTTAATTTATTGTTATAAAC | *pocR* | No RE |
| T7P *eutS* For | CTAATACGACTCACTATAGGGACAAAAAATTGCCACGATGACGGCAG | *eutS* | No RE |
| *eutS* Rev | GACCTGTTTGCCCGGCACAAATTCCTGAATAATG | *eutS* | No RE |

$^a$Italic letters indicate the respective restriction enzyme site in the primer.
$^b$The sequences corresponding to the template plasmids pKD4 or pSUB11 are underlined.
$^c$RE, restriction enzyme for which a site was generated in the primer.

harvested from 1.5 mL of bacterial culture, and they were then centrifuged, washed, resuspended in 800 $\mu$L of TDTT buffer (50 mM Tris-HCl [pH 7.8] and 30 $\mu$M DL-dithiothreitol [DTT]), and sonicated on an ice bath for 3 min with periods of 10 s of sonication and 10 s of rest. The sonicated samples were centrifuged, and the supernatant was used for the protein and activity measurements. For the $\beta$-galactosidase activity, 20 $\mu$L of each soluble extract was added to a 96-well microplate, and this was followed by the addition of 200 $\mu$L of 0.5 mg/mL o-nitrophenyl $\beta$-D-galactopyranoside (ONPG) resuspended in 1× Z buffer (60 mM Na$_2$HPO$_4$, 46 mM NaH$_2$PO$_4$, 10 mM, KCl, 2 mM MgSO$_4$) pH 7. The rate of each reaction was obtained by recording the change in absorbance at 405 nm every 15 s for 5 min by using an ELx808 scanning microplate reader and the Gen5 software package (BioTek). The activities were obtained via interpolation with a standard curve (0 to 5,400 U) that was previously stored in the Gen5 software. The protein concentration of each cell extract was obtained by using a bicinchoninic acid (BCA) protein assay (Pierce). Bovine serum albumin was used as the protein standard. The enzyme activity and protein concentration values were used to calculate the $\beta$-galactosidase specific activity (U/mg).

**Purification of maltose binding protein (MBP)-PocR.** *E. coli* BL21/DE3 carrying the pMAL-PocR was grown in two flasks containing 100 mL of LB with 0.2% glucose at 37°C in a shaken water bath. At an optical density of 0.6, the expression of MBP-PocR was induced by the addition of 1 mM isopropyl-$\beta$-D-thiogalactopyranoside (IPTG). Then, the cultures were incubated overnight at 18°C. Bacterial cells were collected by centrifugation at 8,000 rpm at 4°C. The pellet was washed once with ice-cold column buffer (200 mM Tris-HCl [pH 7.5], 200 mM NaCl, 1 mM EDTA, and 10 mM $\beta$-mercaptoethanol) and resuspended in 30 mL of the same buffer. The bacterial suspension was sonicated in a Soniprep 150 sonicator (Sonics and Materials, Inc.). Bacterial debris was separated by centrifugation at 4°C, and the soluble extract was loaded three times into an amylose column (New England Biolabs) equilibrated with column buffer. The column was then washed with 13 volumes of column buffer. MBP-PocR was eluted with column buffer containing 10 mM maltose (Sigma). The fractions were analyzed by using SDS-12% polyacrylamide gels, and those containing purified MBP-PocR were concentrated in 1 mL of dialysis buffer (20 mM Tris-HCl [pH 8], 40 mM KCl, 1 mM EDTA, 1 mM DTT, and 20% [vol/vol] glycerol) using an Amicon Ultra 50K device (Merck Millipore) at 5,000 × *g* for 20 min. The protein concentration was determined by the Bradford method. Aliquots of the purified proteins were stored at −70°C.

**DNA EMSAs.** DNA fragments containing the intergenic region upstream of *pduA*, *pocR*, and *eutR* were obtained by PCR amplification with the same primer pairs that were used to construct the respective *lacZ* translational fusions. The resulting DNA fragments were purified with a Zymo DNA Clean & Concentrator Kit (Zymo Research). Each PCR product (100 ng) was mixed with increasing concentrations

of purified MBP-PocR in 20 $\mu$L of binding buffer containing 10 mM Tris (pH 8.0), 50 mM KCl, 1 mM DTT, 0.5 mM EDTA, 5% glycerol, and 10 $\mu$g/mL bovine serum albumin (BSA). The binding reaction mixtures were incubated at room temperature for 20 min, mixed with 2 $\mu$L of 5$\times$ DNA-loading dye (0.25% bromophenol blue, 0.25% xylene cyanol FF, 30% glycerol, and 50$\times$ Gel Red [Biotum]), and then analyzed on 6% nondenaturing Tris-borate-EDTA (TBE)-buffered acrylamide gels in 0.5$\times$ TBE buffer at 85 V and room temperature. After electrophoresis, the DNA fragments were visualized by UV light illumination (Bio-Rad Molecular Imager, Gel Doc TM, XR+ Imaging System, USA). The MBP-PocR complex was detected by Western blotting.

**RNA EMSAs.** The electrophoretic mobility shift assays (EMSAs) were performed using published procedures (70). His-tagged CsrA (CsrA-H6) was purified as previously described (71). Note that the sequences of CsrA from *Salmonella* and *E. coli* are identical. RNA was synthesized *in vitro* by using a RNAMaxx Transcription Kit (Agilent Technologies). The PCR fragments that were used as the templates in the transcription reactions contained a T7 promoter and *pocR*, *pduA*, and *eutS* sequences that extended from −91 to +3, −44 to +38, and −47 to +48, relative to the start of the translation, respectively. Gel-purified RNA was 5′-end-labeled with [$\gamma$-$^{32}$P]-ATP (7,000 Ci/mmol). RNA suspended in TE buffer was heated to 90°C for 1 min, and this was followed by slow cooling to room temperature. The binding reaction mixtures (10 $\mu$L) contained 10 mM Tris-HCl (pH 7.5), 10 mM MgCl$_2$, 100 mM KCl, 200 ng/$\mu$L yeast RNA, 0.2 mg/mL BSA, 7.5% glycerol, 20 mM DTT, 0.1 mg/mL xylene cyanol, 0.2 nM (*eutS* and *pduA*) or 0.1 nM (*pocR*) RNA, and various concentrations of purified CsrA-H6. The competition assay mixtures also contained unlabeled competitor RNA. The reaction mixtures were incubated for 30 min at 37°C to allow for CsrA-RNA complex formation. The samples were then fractionated through native 10% polyacrylamide gels using 0.5$\times$ TBE buffer. Radioactive RNA bands were visualized with a Typhoon 9410 phosphorimager (GE Healthcare) and quantified using the ImageQuant 5.2 software package. The apparent equilibrium binding constants ($K_d$) of the CsrA-RNA interaction were calculated as previously described (72).

**Western blotting.** Western blot assays were conducted as previously described (13), with the following modifications. Briefly, 50 $\mu$g of bacterial soluble extracts were subjected to electrophoresis. Monoclonal anti-FLAG M2 (Sigma), anti-MBP (Sigma), and polyclonal anti-GroEL (Pierce) antibodies were used at 1:5,000, 1:3,000, and 1:50,000 dilutions, respectively. Horseradish peroxidase-conjugated secondary anti-mouse or anti-rabbit antibodies (Pierce) were used at 1:10,000 dilutions.

**Statistical analysis.** Statistical significance was analyzed by using the Prism 8 program, version 8.01 (GraphPad Software, San Diego, CA), using either one-way analyses of variance (ANOVA) with Tukey's *post hoc* tests or unpaired Student's *t* tests. A *P* value of <0.05 was considered to be indicative of a statistically significant result.

## ACKNOWLEDGMENTS

This work was supported by grants from the Dirección General de Asuntos del Personal Académico de la UNAM/México (IN206321) to V.H.B. and from the National Institutes of Health (GM059969 and GM098399) to P.B. J.N.-G. was supported by a predoctoral fellowship from CONACYT (679158).

We thank F.J. Santana and M. Fernández-Mora for the technical assistance as well as I. Martínez-Flores for the critical reading of the manuscript.

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
