## [Reviewer comments · Microbiology Spectrum]

Microbiology Spectrum

CsrA positively and directly regulates the expression of the *pdu*, *pocR*, and *eut* genes required for luminal replication of *Salmonella* Typhimurium

Jessica Nava-Galeana, Helen Yakhnin, Paul Babitzke, and Victor Bustamante

Corresponding Author(s): Victor Bustamante, Universidad Nacional Autonoma de Mexico - Campus Morelos

Review Timeline:

Submission Date:	April 10, 2023
Editorial Decision:	May 8, 2023
Revision Received:	May 11, 2023
Accepted:	May 26, 2023

Editor: Fernando Navarro-Garcia

Reviewer(s): The reviewers have opted to remain anonymous.

Transaction Report:

DOI: <https://doi.org/10.1128/spectrum.01516-23>

May 8, 2023

Dr. Victor H. Bustamante
Universidad Nacional Autonoma de Mexico - Campus Morelos
Microbiologia Molecular
Av. Universidad 2001
Col. Chamilpa
Cuernavaca, Morelos 62210
Mexico

Re: Spectrum01516-23 (*CsrA* positively and directly regulates the expression of the *pdu*, *pocR*, and *eut* genes required for luminal replication of *Salmonella* Typhimurium)

Dear Dr. Victor H. Bustamante:

Link Not Available

Sincerely,

Fernando Navarro-Garcia

Journals Department
Reviewer comments:

Reviewer #1 (Comments for the Author):

Nava-Galeana et al. show that the Bar/SirA system regulates the *pdu* and *eut* operons via *CsrA*. This is a relatively straightforward study that adds to our understanding of *CsrA* regulation of virulence and physiology. The experiments are well conceived and the results are convincing. I have only a few minor comments.

1. Abstract - Line 28. This sentence that starts "Furthermore..." needs to be re-structured. As is, it says that the SirA cascade

depends on PocR or UteR. No what you mean.

2. Line 104. You might want to mention that *csrA* mutations are lethal/sick, hence you are not testing in this background.
3. Fig 6 shows regulation in when cells are grown in the presence of substrate/inducer. Although not a big deal, it would be nice to have it side-by-side, with and without, to see the relative effect.
4. The paper would benefit significantly from a model slide explaining the overall results and conclusions, tying it in with Spi1 (and flagellar?) regulation.

Reviewer #3 (Comments for the Author):

The manuscript by Nava-Galeana et al. investigates the regulatory cascades controlling *Salmonella* virulence. Specifically, the look at the genes controlling expression of SPI-1 and those for metabolizing propanediol and ethanolamine. Previous data indicated that the carbon storage Csr system regulating the *pdu* and *eut* genes, but the mechanism of how this occurs was undetermined. The authors use multiple molecular techniques, including construction transcriptional and translational fusions to monitor gene expression assayed in different genetic backgrounds, as well as protein binding assays to both DNA and RNA.

The authors find that CsrA activates the expression of the *pdu* and *eut* genes by directly binding to transcripts, and this expression requires the PocR and EutR regulators, respectively. Overall, the manuscript is well written and the data support the authors' conclusions. The discussion containing current thoughts on *Salmonella* virulence, and how this work contributes, is well received. Comments for improving the submission are included below.

Major concerns:

1. For protein expression of PocR and EutR using FLAG tag constructs in Figure 4 it would be useful to have quantitation using replicates, densitometry and statistical analysis.
2. Figure 5. The authors use EMSAs to show binding of PocR to *eutR* regulatory DNA, and use the lack of binding to *eutR* and *pocR* to serve as a negative control. However, the concentration of protein used is relatively high, in the micromolar range. Because EMSAs are relatively sensitive, often in the nanomolar range for protein binding, how does this affect the conclusion that PocR binds specifically to *eutR*?

Minor concerns:

1. Line 139. The authors state that they have constructed a "*pocR*'-lacZ translational fusion" but the rationale for this construct is not given. The implication is that all other promoter fusion constructs are transcriptional, but some clarity here would be useful.
2. Lines 178 to 180. It is difficult to reconcile this statement with genetic organization of the different loci in Figure 1.
3. Lines 186 to 189. The authors state regulation in the presence of inducer molecules, propanediol, etc., but it is unclear on Figure 6 to which graphs, or bars they are referring. Please clarify.

Staff Comments:

Preparing Revision Guidelines

Please return the manuscript within 60 days; if you cannot complete the modification within this time period, please contact me. If you do not wish to modify the manuscript and prefer to submit it to another journal, please notify me of your decision immediately so that the manuscript may be formally withdrawn from consideration by Microbiology Spectrum.

REPLAY TO COMMENTS BY REVIEWERS / Spectrum01516-23 R1

We are very grateful with the reviewers for their constructive comments that really helped us to improve the strength of our study.

Comments

Reviewer #1 (Comments for the Author):

Nava-Galeana et al. show that the Bar/SirA system regulates the *pdu* and *eut* operons via CsrA. This is a relatively straightforward study that adds to our understanding of CsrA regulation of virulence and physiology. The experiments are well conceived and the results are convincing. I have only a few minor comments.

Thanks for your positive opinion.

1. Abstract - Line 28. This sentence that starts "Furthermore..." needs to be re-structured. As is, it says that the SirA cascade depends on PdcR or UteR. No what you mean.

Thanks for pointing this out. This sentence now reads:

Furthermore, our results show that the SirA-CsrB/CsrC-CsrA regulatory cascade controls the expression of the *pdu* and *eut* genes mediated by PdcR or EutR, the positive AraC-like transcriptional regulators for the *pdu* and *eut* genes, respectively.

2. Line 104. You might want to mention that *csrA* mutations are lethal/sick, hence you are not testing in this background.

This is an important statement, it was included. We appreciate your suggestion.

3. Fig 6 shows regulation in when cells are grown in the presence of substrate/inducer. Although not a big deal, it would be nice to have it side-by-side, with and without, to see the relative effect.

This figure was modified as suggested.

4. The paper would benefit significantly from a model slide explaining the overall results and conclusions, tying it in with Spi1 (and flagellar?) regulation.

This model was included (Fig. 8).

Reviewer #3 (Comments for the Author):

The manuscript by Nava-Galeana et al. investigates the regulatory cascades controlling *Salmonella* virulence. Specifically, the look at the genes controlling

expression of SPI-1 and those for metabolizing propanediol and ethanolamine. Previous data indicated that the carbon storage Csr system regulating the pdu and eut genes, but the mechanism of how this occurs was undetermined. The authors use multiple molecular techniques, including construction transcriptional and translational fusions to monitor gene expression assayed in different genetic backgrounds, as well as protein binding assays to both DNA and RNA.

The authors find that CsrA activates the expression of the pdu and eut genes by directly binding to transcripts, and this expression requires the PocR and EutR regulators, respectively. Overall, the manuscript is well written and the data support the authors' conclusions. The discussion containing current thoughts on Salmonella virulence, and how this work contributes, is well received. Comments for improving the submission are included below.

Thanks for your positive comments.

Major concerns:

1. For protein expression of PocR and EutR using FLAG tag constructs in Figure 4 it would be useful to have quantitation using replicates, densitometry and statistical analysis.

As suggested, we quantified the expression of PocR-FLAG and EutR-FLAG from three different blots that we had done previously. The new data added to Figure 4 further support our conclusion about the positive regulation of PocR and EutR by CsrA.

2. Figure 5. The authors use EMSAs to show binding of PocR to eutR regulatory DNA, and use the lack of binding to eutR and pocR to serve as a negative control. However, the concentration of protein used is relatively high, in the micromolar range. Because EMSAs are relatively sensitive, often in the nanomolar range for protein binding, how does this affect the conclusion that PocR binds specifically to eutR?

Yes, the concentration used in our non-radioactive EMSAs are higher than those used in classical radioactive EMSAs. This difference is due to the high amount of template DNA used in non-radioactive EMSAs to detect DNA bands by ethidium bromide stain, compared with radioactive EMSAs. That is the reason why we include internal and external negative controls in these assays. We have used non-radioactive EMSAs in several studies with good results (Bustamante et al., 2008, PNAS 105:14591-14596; Pérez-Morales et al., 2017, PLoS Pathog 1006497; Banda et al., 2022, J Bacteriol 204: issue 11).

Minor concerns:

1. Line 139. The authors state that they have constructed a "pocR'-lacZ translational fusion" but the rationale for this construct is not given. The implication is that all other promoter fusion constructs are transcriptional, but some clarity here would be useful.

Sorry by the misunderstanding. All the *lacZ* fusions we used are translational because CsrA affect translation, or stability of target mRNAs. We emphasize this in the schematic representation of fusions (Figure 1) and through the manuscript.

2. Lines 178 to 180. It is difficult to reconcile this statement with genetic organization of the different loci in Figure 1.

Thanks for your observation. This statement now reads:

Together, these results indicate that the SirA-CsrB/C-CsrA cascade regulates expression of *pocR* by acting on the transcript generated by the promoter upstream of this gene, and regulates expression of *eutR* by acting on the transcript generated by the promoter upstream of *eutS*.

3. Lines 186 to 189. The authors state regulation in the presence of inducer molecules, propanediol, etc., but it is unclear on Figure 6 to which graphs, or bars they are referring. Please clarify.

For a better understanding, Figure 6 now shows results obtained in the absence and presence of inducers, which is indicated.

May 26, 2023

Dr. Victor H. Bustamante
Universidad Nacional Autonoma de Mexico - Campus Morelos
Microbiologia Molecular
Av. Universidad 2001
Col. Chamilpa
Cuernavaca, Morelos 62210
Mexico

Re: Spectrum01516-23R1 (CsrA positively and directly regulates the expression of the *pdu*, *pocR*, and *eut* genes required for luminal replication of *Salmonella* Typhimurium)

Dear Dr. Victor H. Bustamante:

Your manuscript has been accepted, and I am forwarding it to the ASM Journals Department for publication. You will be notified when your proofs are ready to be viewed.

Sincerely,

Fernando Navarro-Garcia
Editor, Microbiology Spectrum
